# Research on Co-Opetition Mechanism between Pharmaceutical Enterprises and Third-Party Logistics in Drug Distribution of Medical Community

**DOI:** 10.3390/ijerph20010609

**Published:** 2022-12-29

**Authors:** Zhao Li, Tie Xia, Wanzhi Shen, Sheng Chen

**Affiliations:** School of Management, Jiangsu University, Zhenjiang 212013, China

**Keywords:** medical community, drug distribution, co-opetition, third-party logistics

## Abstract

Third-party logistics (3PL) has a relatively perfect distribution system in solving the drug distribution of the medical community and optimizing the distribution efficiency of pharmaceutical enterprises, and it has gradually become an indispensable component of drug distribution. By constructing the co-opetition model of “Pharmaceutical Enterprises—3PL”, this paper explores the game strategy choice between pharmaceutical enterprises and 3PL for the solution of drug distribution under the condition of information asymmetry, and it puts forward some suggestions to improve the competition and cooperation mechanism between pharmaceutical enterprises and 3PL in drug distribution in the medical community.

## 1. Introduction

The construction of the medical community is an important measure in Chinese medical reform and the main mode of rural medical alliance construction. It focuses on exploring the integrated management of the county and village with “county and district public hospitals as the leader, township hospitals as the hub, and hamlet clinics as the basis”, and building a county-level medical service system with county, village, and hamlet linkage. In July 2020, the National Health Commission issued the “Administrative Measures for Medical Consortium (Trial)”, which indicated that the construction of the medical consortium had entered a substantive stage of advancement [1]. It can be seen that the state attaches great importance to the construction of the grassroots medical and health system, which will provide a broader space for the development of the county-level medical community drug distribution market.

As far as the current situation of the development of the medical community model is concerned, rural medical and health institutions, personnel allocation and basic medical security systems are provided at a basic level in grassroots areas, but there are still many problems in drug distribution. For example, the users in grassroots areas are scattered, the drug types are various, the drug distribution frequency is high, and the county transportation network of pharmaceutical enterprises is backward—these cannot meet the drug distribution needs in grassroots areas. At the same time, the medical community has implemented provisions on the centralized bidding procurement of drugs in terms of drug circulation, and it has implemented unified procurement and distribution, which has led to a serious monopoly, a lack of competition mechanisms, and an insufficient supply of some major drugs in the process of distribution of essential drugs. To alleviate the problems of slow drug turnover in the medical community and inconvenient drug use by grassroots people [2], in 2016, the Office of the State Council issued “the Decision on the Second Batch of Cancellation of 152 Administrative Examination and Approval Matters in Central Designated Places” (GF [2016] No. 9) and “the Guiding Opinions on Promoting the Healthy Development of the Pharmaceutical Industry” (GBF [2016] No. 11), eliminating the need for the administrative examination and approval of engagement in third-party drug logistics business; as long as the 3PL meets the standards, it can enter the pharmaceutical distribution. The authorities also stated that they would give full play to the advantages of the postal and express delivery networks of postal enterprises and express enterprises, and improve the drug supply guarantee capability in grassroots and remote areas [3,4]. On 28 May 2019, the National Health Commission and the State Administration of Traditional Chinese Medicine issued “the Notice on Promoting the Construction of a Close County Medical and Health Community” (GWQJH [2019] No. 121), indicating that in areas where conditions permit, the difference between different medical communities in the county should be alleviated, and the unified management and procurement distribution of drug consumables in the county should be explored [5]. On 31 August 2020, the National Health Commission, the National Health Insurance Bureau, and the National Bureau of Traditional Chinese Medicine issued “the Notice on Issuing the Criteria and Monitoring Index System for the Construction of Tight County Medical and Health Communities (for Trial Implementation)”, which proposed that “unified drug management: unified management of drug consumables in the county medical community, unified procurement and distribution, unified payment for goods, unified drug catalog, etc.” should be one of the evaluation criteria for the construction of a tight county medical and health community [6]. On 28 October 2021, the Ministry of Commerce issued “the Guiding Opinions on Promoting the High Quality Development of the Drug Distribution Industry during the ‘Fourteenth Five Year Plan’”, proposing to accelerate the construction of the rural drug distribution network, gradually improve the county, township, and rural three-level drug distribution system, and support drug distribution enterprises to conduct market-oriented cooperation with 3PL, postal services, express delivery, etc. [7].

Therefore, this paper analyzes the drug distribution problems faced by grassroots hospitals under the unique medical community model in China, and it introduces the evolutionary game idea to optimize the drug distribution problems faced in the development of the Chinese medical community. Compared with the existing research, the main contributions of this paper are as follows. (1) The research conducted under the medical community model in China has a distinctive Chinese situational feature, which can promote the coordinated development of Chinese pharmaceutical enterprises and 3PL and provide a guarantee for the effective operation of the medical community’s drug distribution. (2) The introduction of the idea of evolutionary game and the exploration of the rules of strategy selection of both parties in the evolutionary process of competition and cooperation behavior can confirm the feasibility of the competition and cooperation game in the drug distribution of China’s medical community. (3) The research results will help to optimize the competition and cooperation relationship between pharmaceutical enterprises and 3PL enterprises in the drug distribution of the Chinese medical community, innovate and improve the theoretical cognition of drug distribution in the medical community, and optimize the corresponding management practices.

## 2. Literature Review

Co-opetition was first proposed by Nalebuff and Brandenburger, as a business strategy inspired by game theory [8]. The main aspect is building partnerships on a competitive basis. At different stages, a binary relationship of both cooperation and competition is finally formed through mutual guidance, transformation, connection, and dependence between competition and cooperation [9,10]. The main reason for competition and cooperation among enterprises is to reduce costs and increase profits [11]. In recent years, research on co-opetition between supply chains has become increasingly abundant. Gnyawali, D.R. and Park, B (2009) explored the principles and effects of co-opetition between firms and found that co-opetition for large enterprises on the one hand can bring benefits to participating enterprises and promote technological innovation; on the other hand, it is also conducive to driving the formation of competition and cooperation among other small- and medium-sized enterprises [12]. Hu, X (2011) established three models of competition–competition, cooperation–cooperation, and competition–cooperation when there is a core enterprise in the supply chain as the decision-making subject. By comparing the equilibrium solutions, it is found that the results and utility of the co-opetition model are better than those of the purely competitive strategy [13]. Okura, M (2007), based on the research of DucoeNGO and Mahito OKURA, further used the co-opetition model method of game theory to study the co-opetition strategies of various companies in the Japanese insurance market [14]. Guo, S et al. (2019) studied the fashion supply chain, composed of one manufacturer and two competing retailers, and obtained the best green level of the market through competition and cooperation games [15]. Zhao, L et al. (2017) studied the tripartite cooperation–competition game model for the convergence product market and proposed rich nonlinear dynamic behavior to explain the complex relationship between the three participants [16]. Yang, W et al. (2022) studied the co-opetition strategy between the manufacturer of a proprietary component (MPC) and the original equipment manufacturer (OEM), analyzed the optimal cooperation strategy of MPC, and showed that the existence of third-party suppliers was the key factor in their competition and cooperation behavior through the model [17]. Li, X et al. (2021) found that the retailer’s profit was the largest under the Bertrand competition mode through the equilibrium solution by establishing the centralized and decentralized game-theoretic models under the Bertrand competition, Stackelberg competition, and collusion competition modes [18]. Co-opetition behavior is rarely included in research on the pharmaceutical supply chain. Malmir, B (2016) introduced game theory into the pharmaceutical supply chain, considered the optimal strategies of pharmaceutical companies participating in market competition and set up a reward mechanism to promote the execution of high-quality strategies among competing companies [19].

Drug distribution itself has aroused many concerns. Yuan, Z et al. (2022), in order to reduce the risk and cost of drug distribution, based on the idea of big data, built a multicenter location and route optimization model under the dynamic uncertainty of a pharmaceutical logistics company [20]. Campelo, P et al. (2018) used a decomposition method based on mathematical programming to solve the problem of consistent vehicle routing, and they analyzed a pharmaceutical distribution company [21]. Sabouhi, F et al. (2018) designed a resilient supply chain considering quantity discounts and the risk of drug delivery disruptions, using a variety of proactive strategies to increase supply chain resilience and reduce the risk of threats to the supply chain to reduce costs [22]. Lawrence J M et al. (2020) identified potential factors through Bayesian network approach modeling and analysis to prevent disruption of pharmaceutical supply chains under catastrophic weather events [23]. Hamdan, B et al. (2020) constructed a dual-objective robust optimization model to reduce the time and cost of post-disaster blood distribution, while considering transportation disruption [24]. Dewi, EK et al. (2019) designed a drug inventory management system to avoid the waste caused by the expiration and deterioration of drugs in the inventory due to long-term storage, and to reduce the probability of drug shortages, thus ensuring that patients receive their drugs in time [25]. Acosta, A et al. (2019) proposed the types and causes of drug shortage and supply interruption events, and their classification described the methods and relevant laws and regulations for reporting and managing drug shortage events in different countries according to the updated frequency [26]. Sarnola, K et al. (2019) studied the situation of drug shortage in countries with small drug markets (Finland) due to EU policies or single-channel systems, and they solved the problem of Finland’s key drug shortage by adopting mandatory drug reserve measures and measures introduced at the national and transnational levels [27]. The existing research in the field of drug distribution in China focuses on the historical changes of reform policies, the logical relationship between policies, and the re-sorting of some values. However, the roles of stakeholders in the policy implementation process are not given sufficient attention. In recent years, the issue of drug distribution in county-level medical communities has attracted the attention of scholars, inspired by the 3PL access to drug distribution, aiming at the insufficient supply of drugs at the grassroots level, the slow turnover of drugs in some regions, and the difficulty of drug use by the grassroots population. At the same time, pharmaceutical enterprises always focus on R&D and production, ignoring the problems of transportation, distribution, and other circulation links. Marasco, A (2008) believed that companies can use 3PL to reduce their operating costs and focus on their core business [28]. Grabowski H (2002) mainly conducted research in the field of pharmaceutical distribution and improved the distribution efficiency through the development and management of pharmaceutical distribution channels [29]. Jin, Z and Yu, L (2010) found, in their research, that 3PL has obvious comparative advantages. The introduction of 3PL can improve China’s current pharmaceutical logistics level and improve the efficiency of drug distribution and reduce distribution costs [30]. Zhai, Y and Li, Y (2017) found that the key factors of pharmaceutical logistics costs are transportation costs and distribution center locations by establishing an ISM model [31]. Kramer, R et al. (2019) designed a multi-starting-point iterative local search algorithm using multiple neighborhoods to solve the path distribution problem faced by 3PL in drug delivery [32]. El Mokrini, A et al. (2016) took the pharmaceutical industry as an example to evaluate the risks of outsourcing logistics, and they proposed a decision model considering the risks of outsourcing logistics in the pharmaceutical supply chain [33]. The existing research mainly introduces 3PL to share the problem of insufficient drug distribution among pharmaceutical enterprises, and it rarely considers the competitive behavior between pharmaceutical enterprises and 3PL in the drug distribution market.

In view of this, this paper starts from the problem of drug distribution in the county-level medical community, addressing the lack of corresponding research on the co-evolutionary behavior of the full integration of pharmaceutical companies and 3PL under the current background. This paper considers the motivation of pharmaceutical companies and 3PL behavior strategy choice, and the relevant role of stakeholders, in order to build a co-opetition model between pharmaceutical companies and 3PL; it explores the influence of changing the incremental income of cooperation, the distribution ratio of cooperative income, and the incentive parameters on the co-opetition behavior of both parties by analyzing the evolutionary game of their strategic behavior and puts forward corresponding suggestions.

## 3. Model Construction

### Construction of Co-Opetition Model between Pharmaceutical Enterprises and 3PL Enterprises

At the present stage, both pharmaceutical enterprises and 3PL enterprises can undertake the task of drug distribution in the county-level medical community. However, they have different advantages and disadvantages when providing drug delivery services. For pharmaceutical enterprises, with strong professional strength, they have more advantages in the terminal acceptance, storage, and maintenance of pharmaceutical products, but their shortcomings are the small distribution coverage and low distribution efficiency in county areas. For 3PL enterprises, their weakness lies in their poor discrimination ability, while their advantage lies in their wide coverage and low distribution cost. Based on the differences in their drug delivery services, pharmaceutical enterprises and 3PL enterprises have a competitive relationship. At the same time, pharmaceutical enterprises can also cooperate with 3PL enterprises to complete the distribution of pharmaceutical products, so as to achieve the purpose of obtaining the maximum benefit at a low cost and achieve a “win–win” scenario. Therefore, there is a cooperative relationship between the two. Based on the above analysis, this paper intends to discuss the competition and cooperation relationship between pharmaceutical enterprises and 3PL enterprises. Since neither party knows the strategy of the other party in advance, it has information asymmetry. Both are profit-seeking and conform to the assumption of bounded rationality. At the same time, both of them have the ability to imitate and learn, and they adjust their strategies dynamically according to the previous game situation. Based on the above analysis, the following assumptions are made.

**Assumption 1.** 
*The strategy set of pharmaceutical enterprises is*

$\left\{a_{1},a_{2}\right\}$

*= {cooperation, competition}, and the strategy set of 3PL enterprises is*

$\left\{b_{1},b_{2}\right\}$

*= {cooperation, competition}. The probability that pharmaceutical enterprises choose cooperation is*

x

*, and the probability that pharmaceutical enterprises choose competition is*

(1−x)

*. The probability that 3PL enterprises select cooperation is y, and the probability of selecting competition is (1−y).*


**Assumption 2.** 
*The basic returns of pharmaceutical enterprises and 3PL enterprises in the game are*

$P_{m}$

*and*

$P_{l}$

*, respectively. When both parties cooperate, the quality of the drug delivery service provided is higher, and both parties can obtain the excess revenue*

S

*brought by cooperation. In this case, the maintenance cost of cooperation is*

e

*, and the share ratio of both parties is*

r

*and*

1−r

*, respectively.*


**Assumption 3.** 
*When a pharmaceutical enterprise is combined with a 3PL enterprise strategy for {competition, cooperation}, the benefits of the competition strategy for the pharmaceutical enterprise as*

$a_{m}$

*need to be paid to the 3PL logistics enterprise as the cost of default for μ.*


At this point, the 3PL enterprise shows full cooperation in maintaining costs, and the 3PL enterprise faces a loss of $\eta_{l}$.

**Assumption 4.** 
*When the strategy combination of pharmaceutical enterprises and 3PL enterprises is {cooperation, competition}, the revenue brought by the competitive strategy to the 3PL enterprise is*

$a_{l}$

*, and the breach cost to be paid to the pharmaceutical enterprise is μ. In this case, the pharmaceutical enterprise bears all the cooperation maintenance costs, and the loss faced by the pharmaceutical enterprise is $\eta_{m}$.*


**Assumption 5.** 
*Assuming that the transportation efficiency of 3PL enterprises is positively correlated with their cooperative attitude, when pharmaceutical enterprises choose to cooperate, they will give rewards and punishments according to the behavior of 3PL enterprises. If 3PL enterprises select cooperation at this time, they would obtain incentives from pharmaceutical companies for*

$\delta_{1}R$

*, where R is the incentive cost of pharmaceutical enterprises, and*

$\delta_{1}$

*is the excitation coefficient; the higher*

$\delta_{1}$

*, the higher the incentive degree of pharmaceutical enterprises. Similarly, if a 3PL enterprise chooses to compete, it will be constrained by the pharmaceutical enterprise’s*

$\delta_{2}T$

*, where T is the constraint cost of the 3PL enterprise. The larger*

$\delta_{2}$

*is, the more restrictive pharmaceutical enterprises are towards 3PL enterprises. When pharmaceutical enterprises choose to compete, they will not display incentive and constraint behavior towards 3PL enterprises.*


Therefore, the definition of each parameter is as described in Table 1.

According to the above assumptions and Table 1, the revenue matrix of the competition and cooperation game between pharmaceutical enterprises and 3PL county-level pharmaceutical product distribution is as shown in Table 2.

According to the actual situation, the premise of cooperation is that the excess income obtained by cooperation is greater than the maintenance cost of cooperation, namely s>e, and the benefit of cooperation is greater than the benefit of competition, namely $r\left(s-e\right)>a_{m}\;$and $\left(1-r\right)\left(s-e\right)>a_{l}$. The liquidated damages will not exceed the losses faced by the two when they adopt the cooperation strategy, namely $\mu <e+\eta_{l}$, $\mu <e+\eta_{m}$.

According to Table 2, the expected revenue $E_{A1}$ of pharmaceutical enterprises adopting a cooperative strategy is
(1)$$E_{A1}=y\left[P_{m}+r\left(s-e\right)-\delta_{1}R\right]+\left(1-y\right)\left[P_{m}+\mu -e-\eta_{m}+\delta_{2}T\right]\;\;\;\;\;\;$$

The expected revenue $E_{A2}$ of pharmaceutical enterprises adopting a competitive strategy is
(2)$$E_{A2}=y\left(P_{m}+a_{m}-\mu \right)+\left(1-y\right)\left(P_{m}\right)\;\;\;$$

According to (1) and (2), the expected revenue $E_{A}$ of pharmaceutical enterprises adopting mixed strategies is
(3)$$E_{A}=xE_{A1}+\left(1-x\right)E_{A2}\;\;$$

The expected revenue $E_{B1}$ of a 3PL enterprise adopting a cooperation strategy is
(4)$$E_{B1}=x\left[P_{l}+\left(1-r\right)\left(s-e\right)+\delta_{1}R\right]+\left(1-x\right)\left(P_{l}+\mu -e-\eta_{l}\right)\;\;\;\;\;\;\;\;$$

The expected revenue $E_{B2}$ of a 3PL enterprise adopting a competitive strategy is
(5)$$E_{B2}=x\left(P_{l}+a_{l}-\mu -\delta_{2}T\right)+\left(1-x\right)\left(P_{l}\right)\;$$

According to (4) and (5), it can be seen that the expected revenue $E_{B}$ of a 3PL enterprise adopting a mixed strategy is
(6)$$E_{B}=yE_{B1}+\left(1-y\right)E_{B2}\;\;\;\;\;\;\;\;\;\;\;\;$$

## 4. Model Analysis

### 4.1. Analysis of the Evolution Stability of Pharmaceutical Enterprise Strategy

According to Equations (1) and (3), the replication dynamic equation of the cooperative strategy adopted by pharmaceutical enterprises is
(7)$$\begin{array}{cc} F\left(x\right) & =x\left(1-x\right)\left(E_{A1}-E_{A}\right)\\ & =x\left(x-1\right)\left[\left(a_{m}-e-n_{m}+\delta_{1}R+\delta_{2}T+\left(e-s\right)r\right)y+e-\mu +n_{m}-\delta_{2}T\right]\;\;\; \end{array}$$

Taking the derivative of F(x) yields
(8)$$F^{\prime }\left(x\right)=\left(2x-1\right)\left[\left(a_{m}-e-n_{m}+\delta_{1}R+\delta_{2}T+\left(e-s\right)r\right)y+e-\mu +n_{m}-\delta_{2}T\right]$$

If F(x)=0, two stable states, *x* = 0 and *x* = 1, of the pharmaceutical enterprise system can be solved. Thus, the strategy evolution of pharmaceutical enterprises can be divided into the following three situations:
(1)When $y=\frac{\mu +\delta_{2}T-e-n_{m}}{a_{m}-e-n_{m}+\delta_{1}R+\delta_{2}T+\left(e-s\right)r}$, F(x)=0, at this point, the system is in a stable state for any value of x.(2)When $y>\frac{\mu +\delta_{2}T-e-n_{m}}{a_{m}-e-n_{m}+\delta_{1}R+\delta_{2}T+\left(e-s\right)r}$, $\left(a_{m}-e-n_{m}+\delta_{1}R+\delta_{2}T+\left(e-s\right)r\right)y+e-\mu +n_{m}-\delta_{2}T<0$; therefore, $F^{\prime }\left(1\right)<0,\;F^{\prime }\left(0\right)>0$. In this case, x=1 is an evolutionarily stable strategy—that is, the pharmaceutical enterprises eventually evolve to display cooperative behavior.(3)When $y<\frac{\mu +\delta_{2}T-e-n_{m}}{a_{m}-e-n_{m}+\delta_{1}R+\delta_{2}T+\left(e-s\right)r}$, $\left(a_{m}-e-n_{m}+\delta_{1}R+\delta_{2}T+\left(e-s\right)r\right)y+e-\mu +n_{m}-\delta_{2}T>0$; therefore, $F^{\prime }\left(1\right)>0,\;F^{\prime }\left(0\right)<0$. At this time, x=0 is an evolutionarily stable strategy—that is, the pharmaceutical enterprise will eventually evolve to display competitive behavior.

### 4.2. Evolution Stability Analysis of 3PL Enterprise Strategy

According to Equations (4) and (6), the replication dynamic equation of the cooperation strategy adopted by 3PL enterprises is
(9)$$\begin{array}{cc} F\left(y\right) & =y\left(1-y\right)\left(E_{B1}-E_{B}\right)\\ & =y\left(1-y\right)\left[\left(e-a_{l}+n_{l}+\delta_{1}R-\delta_{2}T+\left(e-s\right)\left(r-1\right)\right)x+\mu -e-n_{l}\right] \end{array}$$

Taking the derivative of F(y) yields
(10)$$F^{\prime }\left(y\right)=\left(1-2y\right)\left[\left(e-a_{l}+n_{l}+\delta_{1}R-\delta_{2}T+\left(e-s\right)\left(r-1\right)\right)x+\mu -e-n_{l}\right]\;\;\;\;\;\;\;$$

If F(y)=0, two stable states, y=0 and y=1, of the 3PL enterprise system can be solved. Thus, the strategy evolution of the 3PL enterprise can be divided into the following three situations:(1)When $x=\frac{e+n_{l}-\mu }{e-a_{l}+n_{l}+\delta_{1}R-\delta_{2}T+\left(e-s\right)\left(r-1\right)},F\left(y\right)=0$. At this point, regardless of the value that *y* takes, the system is stable.(2)When $x>\frac{e+n_{l}-\mu }{e-a_{l}+n_{l}+\delta_{1}R-\delta_{2}T+\left(e-s\right)\left(r-1\right)},\left(e-a_{l}+n_{l}+\delta_{1}R-\delta_{2}T+\left(e-s\right)\left(r-1\right)\right)x+\mu -e-n_{l}>0;$ therefore, $F^{\prime }\left(1\right)<0,F^{\prime }\left(0\right)>0$. In this case, *y* = 1 is the evolutionarily stable strategy, and the 3PL enterprise finally evolves to display cooperative behavior.(3)When $x<\frac{e+n_{l}-\mu }{e-a_{l}+n_{l}+\delta_{1}R-\delta_{2}T+\left(e-s\right)\left(r-1\right)},\left(e-a_{l}+n_{l}+\delta_{1}R-\delta_{2}T+\left(e-s\right)\left(r-1\right)\right)x+\mu -e-n_{l}<0$; therefore, $F^{\prime }\left(1\right)>0,F^{\prime }\left(0\right)<0$. In this case, *y* = 0 is the evolutionarily stable strategy, and the 3PL enterprise finally evolves to display competitive behavior.

### 4.3. Evolutionary Stability Analysis of the System

Considering that there is strategic interaction and dependence in the co-opetition between pharmaceutical companies and 3PL companies, either side of the game will change according to a change in the other party’s strategy. Therefore, it is more appropriate to systematically study the stability of strategy selection of pharmaceutical enterprises and 3PL enterprises. As Formulas (7) and (9) constitute the replicator dynamics equation system, let the replicator dynamics equation system be 0. The solvability for five equilibrium points in the system is $O\left(0,\;0\right),A\left(1,\;0\right),B\left(0,\;1\right),C\left(1,\;1\right),D\left(x_{0},\;y_{0}\right)$. Among them, $x_{0}=\frac{e+n_{l}-\mu }{e-a_{l}+n_{l}+\delta_{1}R-\delta_{2}T+\left(e-s\right)\left(r-1\right)}$, and $0<x^{*}<1,\;y_{0}=\frac{\mu +\delta_{2}T-e-n_{m}}{a_{m}-e-n_{m}+\delta_{1}R+\delta_{2}T+\left(e-s\right)r}$, and 0 $<y^{*}<1$.

The evolutionary stability (ESS) of the replicated dynamic system can be evaluated using the Jacobian matrix’s determinant and the trace of the Jacobian matrix [34]. The Jacobian matrix *J* can be obtained from the replicator dynamics equation system—see Equation (11). The determinant and trace of matrix *J* are *det*(*J*) and *tr*(*J*), respectively; if the equilibrium point satisfies *det*(*J*) > 0, *tr*(*J*) < 0, the equilibrium point is an evolutionarily stable strategy for the replicated dynamic system.
(11)$$J=\left[\begin{array}{cc} \frac{\partial F\left(x\right)}{\partial x} & \frac{\partial F\left(x\right)}{\partial y}\\ \frac{\partial F\left(y\right)}{\partial x} & \frac{\partial F\left(y\right)}{\partial y} \end{array}\right]$$

Among them,
$$\frac{\partial F\left(x\right)}{\partial x}=\left(2x-1\right)\left[\left(a_{m}-e-n_{m}+\delta_{1}R+\delta_{2}T+\left(e-s\right)r\right)y+e-\mu +n_{m}-\delta_{2}T\right]\;\frac{\partial F\left(x\right)}{\partial y}=x\left(x-1\right)\left(a_{m}-e-n_{m}+\delta_{1}R+\delta_{2}T+\left(e-s\right)r\right)\;\frac{\partial F\left(y\right)}{\partial x}=y\left(1-y\right)\left(e-a_{l}+n_{l}+\delta_{1}R-\delta_{2}T+\left(e-s\right)\left(r-1\right)\right)\;\frac{\partial F\left(y\right)}{\partial y}=\left(1-2y\right)\left[\left(e-a_{l}+n_{l}+\delta_{1}R-\delta_{2}T+\left(e-s\right)\left(r-1\right)\right)x+\mu -e-n_{l}\right]$$

Therefore, Table 3 displays the determinant and trace for each equilibrium point.

According to Table 3, there are three situations that comprise (cooperation, cooperation) the desired evolutionary stabilization strategy, as follows.

Case 1: If the conditions $\mu -a_{m}-\delta_{1}R-r\left(e-s\right)>0$ and $\mu -a_{l}+\delta_{1}R-\delta_{2}T+\left(e-s\right)\left(r-1\right)>0$ are satisfied, then the evolutionary game system exists (1, 1) as an evolutionarily stable strategy. At this time, both pharmaceutical companies and 3PL companies choose cooperative behavior.

Case 2: If the conditions $a_{m}-e-n_{m}+\delta_{1}R+\delta_{2}T+\left(e-s\right)r>0$, $e-\mu +n_{m}-\delta_{2}T<0$, $e-a_{l}+n_{l}+\delta_{1}R-\delta_{2}T+\left(e-s\right)\left(r-1\right)>0$, $\mu -e-n_{l}<0$ are satisfied, there are two evolutionarily stable points in the system, namely (0, 0) and (1, 1). At this time, pharmaceutical companies and 3PL companies either choose to cooperate or both choose to compete.

According to Table 3, only the points *F*1 (0, 0) and *F*1 (1, 1) satisfy *Tr*(*J*) < 0, *Det*(*J*) > 0, so the stable points of the system are (0, 0) and (1, 1); point (1.0), (0, 1) is an unstable point, and point $\left(x^{*},y^{*}\right)$ is a saddle point. The phase diagram is shown in Figure 1.

Figure 1 clearly shows that the evolutionary game path and the results of the co-opetition of the drug distribution of the county medical community depend on the initial willingness of the pharmaceutical companies and 3PL companies and the location of the saddle point F5 (*x*^*, *y*^*). When the saddle point falls within the A region (namely *x* > *x**, *y* > *y**), the system evolves to point (1, 1); when the saddle point falls within the C region (namely *x* < *x**, *y* < *y**), the system evolves to the point (0, 0); and when the saddle point falls within the B or D region, the system may evolve to (0, 0) or (1, 1); at this time, the system’s stability point is uncertain.

Case 3: If the conditions $a_{m}-e-n_{m}+\delta_{1}R+\delta_{2}T+\left(e-s\right)r<0$, $e-\mu +n_{m}-\delta_{2}T>0$, $e-a_{l}+n_{l}+\delta_{1}R-\delta_{2}T+\left(e-s\right)\left(r-1\right)>0$, $\mu -e-n_{l}<0$ are satisfied, there is no evolutionarily stable strategy in the system; at this time, the strategy selection of pharmaceutical companies and 3PL companies has strong uncertainty.

On the whole, the “win–win” evolutionary stabilization strategy (1, 1) can be realized in case 1; it can be realized by controlling the size and initial value of each influencing factor in case 2, while there is no evolutionary stabilization strategy in case 3. Therefore, this paper will continue to analyze and discuss the stability of control case 3.

## 5. Numerical Simulation

To intuitively reveal the evolution paths of pharmaceutical enterprises and 3PL enterprises and the sensitivity of each important factor, Matlab is used for numerical simulation. According to the actual situation, it is assumed that the values of each parameter are as follows: $P_{m}=400$, $P_{l}=300$, s=150, r=0.5, $a_{m}=90$, $a_{l}=95$, $\eta_{m}=80$, $\eta_{l}=60$, e=50,  μ=90, $\delta_{1}=1.5$, R=20, $\delta_{2}=2$, and T=10. The start time of the simulation is 0, the end time of the simulation is 5, and the simulation unit is not set in detail.

### 5.1. Initial Evolution Path Analysis

First, based on the initial parameters, the numerical evolution diagram of the pharmaceutical enterprise and 3PL enterprise is obtained through Matlab numerical simulation, as shown in Figure 2. At this time, the replicator dynamics equation of the pharmaceutical enterprise is *F*(*x*) = *x*(1 − *x*)(40*y* − 20). When *y* = 0.5, *F*(*x*) = 0, so *y* = 0.5 is the critical value. It can be seen from Figure 2a that when *y* > 0.5, the function image of the pharmaceutical enterprise is a convex function, *F*′ (1) < 0; at this time, *x* = 1 is the evolutionarily stable strategy—that is, the pharmaceutical enterprise finally chooses the cooperation strategy. When *y* < 0.5, the function image of the pharmaceutical enterprise is a concave function, *F*′ (0) < 0; at this time, *x* = 0 is the evolutionarily stable strategy—that is, the pharmaceutical enterprise finally chooses the competitive strategy. The replicator dynamics equation of the 3PL enterprise is *F*(*y*) = *y*(1 − *y*)(75*x* − 20), so *x* = 0.27 is the critical value. Similarly, in Figure 2b, it can be seen that when *x* > 0.27, the image of the 3PL enterprise is a convex function—that is, the 3PL enterprise finally chooses the cooperative behavior. Moreover, if *x* < 0.27, the 3PL enterprise finally chooses the competitive behavior.

To summarize, in terms of a single subject, if one party adopts cooperation, the other party will eventually evolve to display cooperative behavior in the process of repeated games, forming a “win–win” situation. If one party intends to use competition to increase its gains, both parties will eventually evolve to display competitive behavior—that is, the final evolutionarily stable strategy is (cooperation, cooperation) or (competition, competition).

The numerical evolution diagram of the replicator dynamics equation of a single subject was discussed above. Next, the system evolution path of the relationship between pharmaceutical companies and 3PL companies in co-opetition is simulated through numerical simulation. The above initial parameters meet the judgment conditions $a_{m}-e-n_{m}+\delta_{1}R+\delta_{2}T+\left(e-s\right)r>0$, $e-\mu +n_{m}-\delta_{2}T<0$, $e-a_{l}+n_{l}+\delta_{1}R-\delta_{2}T+\left(e-s\right)\left(r-1\right)>0$, $\mu -e-n_{l}<0$. For the second case, the initial evolution path of the system is as shown in Figure 3. As can be seen from Figure 3, the results of the theoretical analysis and numerical simulation are completely consistent, and the two evolution directions of the whole system are (cooperation, cooperation) and (competition, competition), respectively. Under different initial parameter conditions, the evolution direction is only these two ESSs, and the system eventually evolves to an equilibrium point (0, 0) or (1, 1) depending on the initial situation of its saddle point (0.27, 0.5).

### 5.2. Sensitivity Analysis of Important Parameters of Evolutionary Stability of Pharmaceutical Enterprises and 3PL Enterprises

In the drug distribution business of the county medical community, the strategic choices of pharmaceutical enterprises and 3PL enterprises are obviously affected by the incremental revenue *S* of cooperation between the two parties, income distribution ratio *r*, indemnity *μ* and incentive parameters $\delta_{1}$, $\delta_{2}$, so it is necessary to perform a sensitivity analysis on it.

While keeping the other parameters unchanged, the incremental benefits of adopting cooperative behavior are reduced to 100 and 50, respectively, and the evolutionary results are as shown in Figure 4. As can be seen from Figure 4, with the decrease in incremental revenue from cooperation, both pharmaceutical enterprises and third-party logistics institutions eventually evolve to display competitive behavior. For pharmaceutical enterprises, the incremental benefit of cooperation decreases, and the expected benefit of adopting cooperative behavior decreases. In the process of repeated games, pharmaceutical enterprises find that competition is the dominant strategy, so they will change their cooperation strategy and choose competitive behavior; for the same reason, for 3PL enterprises, competition will also make them more profitable. At the same time, we can also find that the pharmaceutical enterprise is more sensitive to the parameter of cooperative incremental benefits, and it evolves more rapidly towards a competitive strategy.

Under the premise that the rest of the parameters remain unchanged, when changing the income distribution ratio to 0.1, 0.9, respectively, the evolution results are as shown in Figure 5. Figure 5 illustrates this point: for pharmaceutical companies, the increase in the income distribution ratio will cause the behavior of pharmaceutical companies to change greatly. When the income distribution ratio increases to 0.5, the pharmaceutical companies finally choose cooperative behavior, and continuing to increase the income distribution ratio will speed up the rate at which companies converge towards cooperative behavior. For 3PL companies, the proportion of revenue distribution increases, and their behavior is similar to that of pharmaceutical companies. It first evolves into cooperative behavior, and then the rate of convergence to cooperative behavior decreases. This shows that the income distribution ratio is not as high as possible, but, for the entire system, only when the income distribution is moderate can a “win–win cooperation” situation be achieved. If the revenue distribution ratio is too unbalanced, the final evolution direction of the system can only be competition. For example, at this stage, most pharmaceutical enterprises in China still provide drug distribution on a self-operated basis, but for grass-roots areas, the logistics cost is still high, customer service is at a low level, and the future market prospects are not optimistic. Pharmaceutical enterprises have begun to choose a powerful 3PL for logistics trusteeship; the trusteeship parties agree on the logistics cost through agreement and entrust it to 3PL for trusteeship through the personnel or equipment trusteeship mode. The trusteeship parties jointly agree on the logistics cost, and the difference between the agreed logistics cost and the actual logistics cost is regarded as the profit of 3PL.

On the basis of the remaining parameters remaining unchanged, when changing the number of liquidated damages to 180 and 300, respectively, the evolution results are as shown in Figure 6. It can be seen from Figure 6 that the increase in liquidated damages increases the rate at which pharmaceutical companies and 3PL companies converge towards cooperative behavior; however, in contrast, 3PL companies are more sensitive to the parameter of liquidated damages.

Under the premise that the other parameters remain unchanged, when changing the excitation coefficient *δ*1 to 3 and 6, respectively, the evolution results are as shown in Figure 7. Figure 7 shows that the incentive of pharmaceutical companies towards the cooperative behavior of 3PL enterprises is a key factor affecting their cooperation; with the increase in the incentive parameter, the rate of 3PL enterprises first converging towards competitive behavior slows down, and then the incentive parameter continues to increase, and the 3PL company finally chooses cooperative behavior. However, excessive incentives are not conducive to the performance of pharmaceutical companies. Changing the punishment parameter will have similar effects, which will not be detailed here, but the difference is that a change in the penalty parameter has a positive incentive effect on the cooperation between the two.

In fact, we can simulate the sensitivity of any parameter numerically, and the simulation results are consistent with the theoretical analysis; however, due to space limitations, we omit them here.

## 6. Evolutionary System Optimization of Pharmaceutical Enterprises and 3PL Enterprises

According to the theoretical analysis, there is an evolutionarily stable strategy (1, 1) in case 1 of the evolutionary system, while the equilibrium of (1, 1) can be achieved in case 2 by adjusting the parameter values. Meanwhile, in case 3, there is strong uncertainty between pharmaceutical enterprises and 3PL enterprises in their strategic choices, and there is no evolutionarily stable strategy. We visualize the behavior of pharmaceutical enterprises and 3PL enterprises in case 3, and the specific parameters are set as follows: $P_{m}=400,\;\;P_{l}=300$, s=150, $r=0.5,\;\;a_{m}=90,a_{l}=130,\eta_{m}=80,\eta_{l}=60,e=50,\mu =90,\;\delta_{1}=1.5,R=40,\;\;\delta_{2}=2,T=25$. The evolution result is shown in Figure 8.

It can be seen from Figure 8 that the strategy choices of pharmaceutical enterprises and 3PL enterprises have strong volatility, and there is no evolutionarily stable strategy. A premium and penalty mechanism is an important means to regulate the behavior of the subject [35]; therefore, it can be considered to control the volatility between pharmaceutical enterprises and 3PL enterprises from the perspective of the premium and penalty mechanism.

### 6.1. Static Premium and Penalty Mechanism

The fourth summary demonstrates that the static premium and penalty mechanism actually alters the fixed value’s size to alter the rate of evolution; the effectiveness of controlling the fluctuation stability is not known, so we adjust the reward parameter to 3 and the penalty parameter to 2.5, respectively. The evolution results are shown in Figure 9 and Figure 10.

It can be seen from Figure 9a that the increase in the incentive parameter does not render the evolutionary game system stable, but changes the peak fluctuation of pharmaceutical enterprises and 3PL enterprises; with the increase in evolutionary time, the strategic choices of 3PL enterprises and pharmaceutical enterprises tend to be almost synchronous—that is, pharmaceutical enterprises tend to choose cooperation, so 3PL enterprises tend to choose cooperation, and when pharmaceutical enterprises tend to choose competition, 3PL enterprises also tend to choose competition. Similarly, it can be seen from Figure 9b that under the change in the punishment parameter, the system is also less able to reach a stable state, but evolves with the fluctuation in the two parties’ strategy choices; however, it can be seen that the peak fluctuation of the punishment mechanism is higher than that of the reward mechanism—that is, the punishment mechanism is more conducive to regulating the behavior of pharmaceutical enterprises and 3PL enterprises than the reward mechanism, but neither of them makes the system stable. It can be seen from Figure 10a,b that Figure 9a,b are special cases of Figure 10 with initial values of [0.6, 0.3], while changes in the initial values of pharmaceutical enterprises and 3PL enterprises at (0, 1) do not cause the behaviors of the two agents to become evolutionary stability strategies. Moreover, changes in different initial values only change the position of the saddle point. On the other hand, it can be seen from the comparison between Figure 9 and Figure 10 that the mechanism of action of the static premium and penalty mechanism is simply to change the magnitude of the fixed value, but it does not control the stability of the system. This is because, with the co-opetition between pharmaceutical enterprises and 3PL enterprises, the strategic choices of both parties are not static but dynamically adjusted. The static premium and penalty mechanism will play a role in controlling stability in a short period of time, but, in the long-term game process, the static premium and penalty mechanism loses its effect with the dynamic adjustment of the strategy, so this paper introduces the dynamic premium and penalty mechanism to control the stability of the system.

### 6.2. Dynamic Premium and Penalty Mechanism

Considering the insufficiency of the static premium and penalty mechanism, this paper introduces the dynamic premium and penalty mechanism to control the stability of the system, and it links the strength of rewards and punishments with the incentive willingness (degree of cooperation) of pharmaceutical enterprises, namely
(12)$$\delta_{1}=\{\begin{array}{c} \delta_{1}\;\;\;\;\;\;\;\;\;\;\;\;\;\;\;\;\;\;\;\;\;\;\;x=1\\ \alpha x\delta_{1}\;\;\;\;\;\;\;\;\;\;\;\;\;\;\;\;\;\;0\leq x<1 \end{array}$$

Among them, *α* is the dynamic reward coefficient, and 0 < *α* ≤ 1. We take α as 1, so the evolution paths of pharmaceutical enterprises and 3PL enterprises under the dynamic premium and penalty mechanism are as shown in Figure 11.

It can be seen from Figure 11 that under the dynamic incentive mechanism, the behavior of pharmaceutical enterprises and 3PL enterprises is no longer in a state of fluctuation, but they reach an evolutionarily stable strategy at the mixed equilibrium point in a relatively short time. Especially for 3PL enterprises, the dynamic incentive mechanism implements incentives according to their cooperation degree, causing third-party logistics enterprises to choose cooperation as an evolutionarily stable strategy; the pharmaceutical enterprises themselves are at the peak of their initial willingness to cooperate, so the dynamic incentive mechanism effectively controls the stability of the evolutionary system. On the other hand, it is not difficult to see that the change in initial value affects the rate of the evolutionary stability strategy of pharmaceutical enterprises and 3PL enterprises, but it does not change the state of evolutionary stability—that is, under different initial values, the strategy choice of pharmaceutical enterprises and 3PL enterprises eventually evolves into the same mixed strategy. Therefore, the system under the dynamic reward and punishment mechanism is no longer in a fluctuating and unstable state, but in a mixed strategy. Therefore, it can be seen that the dynamic reward and punishment mechanism is the stability control strategy of the system. For example, at this stage, the reward and punishment of pharmaceutical enterprises on 3PL reflects the logistics service management; the 3PL supervises and checks the acceptance, maintenance, storage, receiving, and distribution management according to the relevant provisions of the national GSP. To ensure the timely delivery of goods delivered by pharmaceutical enterprises, pharmaceutical enterprises will assess the logistics services of the entrusted party at any time, and the assessment results will be directly reflected in the service rate.

## 7. Conclusions

In the context of promoting the healthy and orderly development of the county medical community, this paper constructs a competition and cooperation model between pharmaceutical enterprises and 3PL enterprises to address the drug distribution problem of the county medical community, and it analyzes the evolution paths and influencing factors between the two parties. The initial evolution paths and the sensitivity of important parameters of pharmaceutical enterprises and 3PL enterprises are simulated by Matlab numerical simulation, and then, considering that the static premium and penalty mechanism cannot control the stability of the system, the dynamic premium and penalty mechanism is introduced to optimize the system, and the following conclusions are drawn.(1)The cooperation and stability strategy of the drug distribution system of the county medical community is positively correlated with the incremental income of the cooperation between the two parties, liquidated damages and the degree of incentive and restraint. For pharmaceutical enterprises, cooperative incremental benefits are more sensitive, while 3PL enterprises are more sensitive to liquidated damages.(2)The income distribution ratio needs to be set reasonably in order to achieve a “win–win cooperation” situation. Values that are too high or too low will cause competition between pharmaceutical enterprises and 3PL enterprises.(3)Numerical simulations intuitively reveal the different evolution paths of pharmaceutical enterprises and 3PL enterprises and further verify the correctness of the theoretical analysis.(4)The static premium and penalty mechanism fails to dynamically adjust according to the behaviors of pharmaceutical enterprises and 3PL enterprises, thus failing to provide a stability control strategy for the system, while the dynamic premium and penalty mechanism links the behavior of the subject with the intensity of rewards and punishments and effectively controls the system’s stability.

## Figures and Tables

**Figure 1 ijerph-20-00609-f001:**
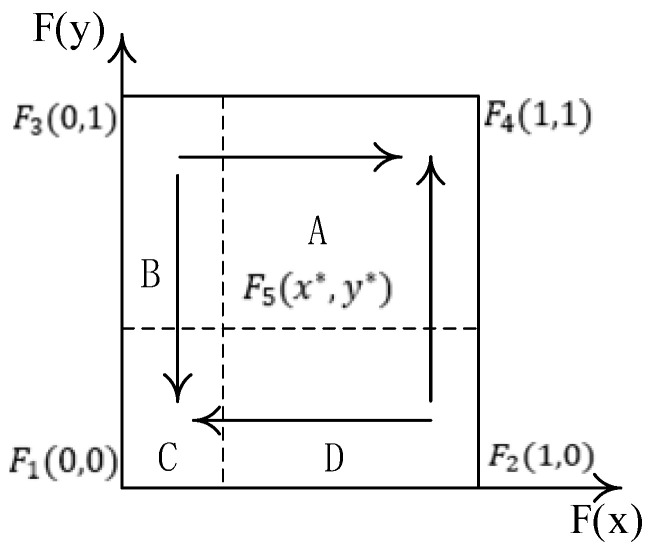
The phase diagram of the co-opetition of drug distribution in the county medical community.

**Figure 2 ijerph-20-00609-f002:**
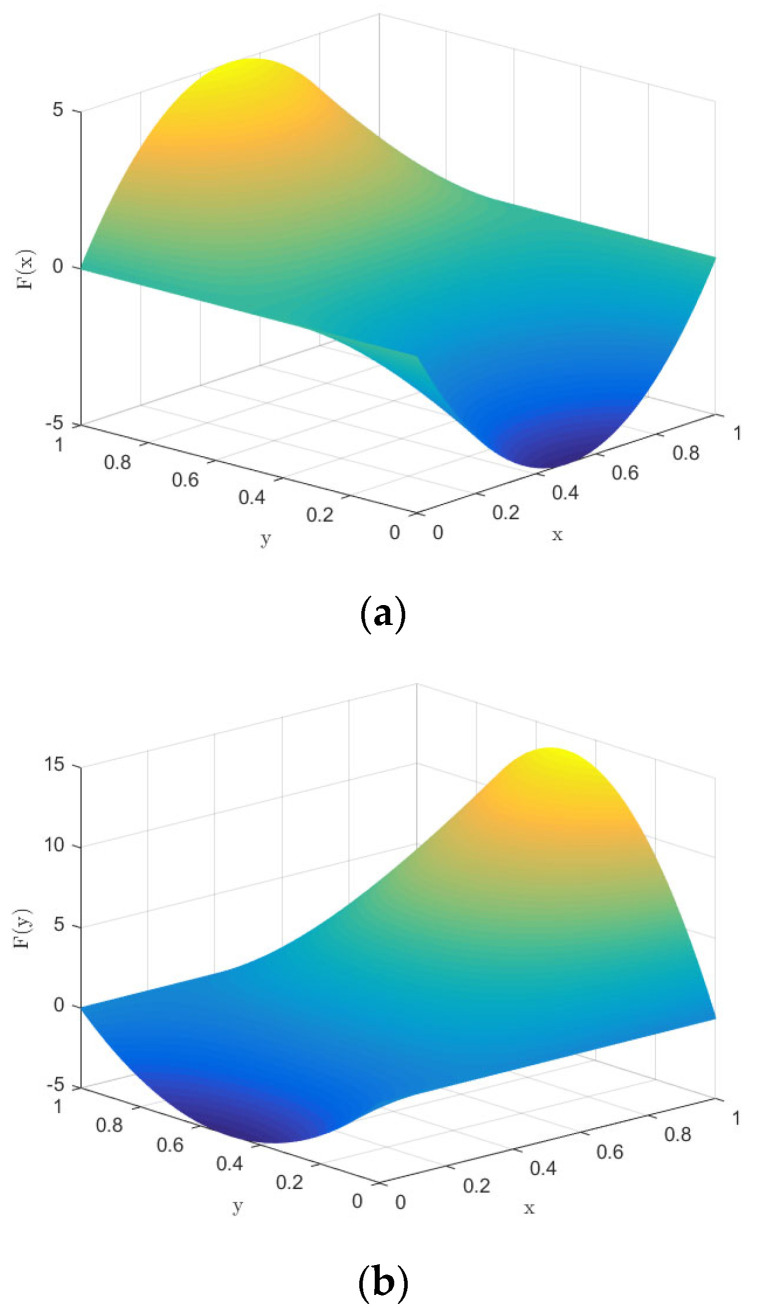
Numerical evolution diagram of pharmaceutical enterprises (**a**) and 3PL enterprises (**b**).

**Figure 3 ijerph-20-00609-f003:**
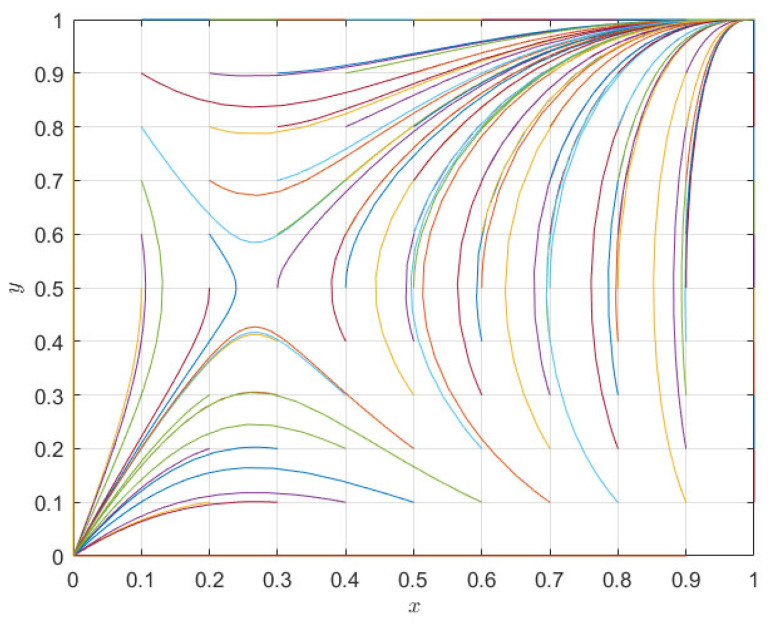
System evolution path diagram of pharmaceutical enterprises and 3PL institutions.

**Figure 4 ijerph-20-00609-f004:**
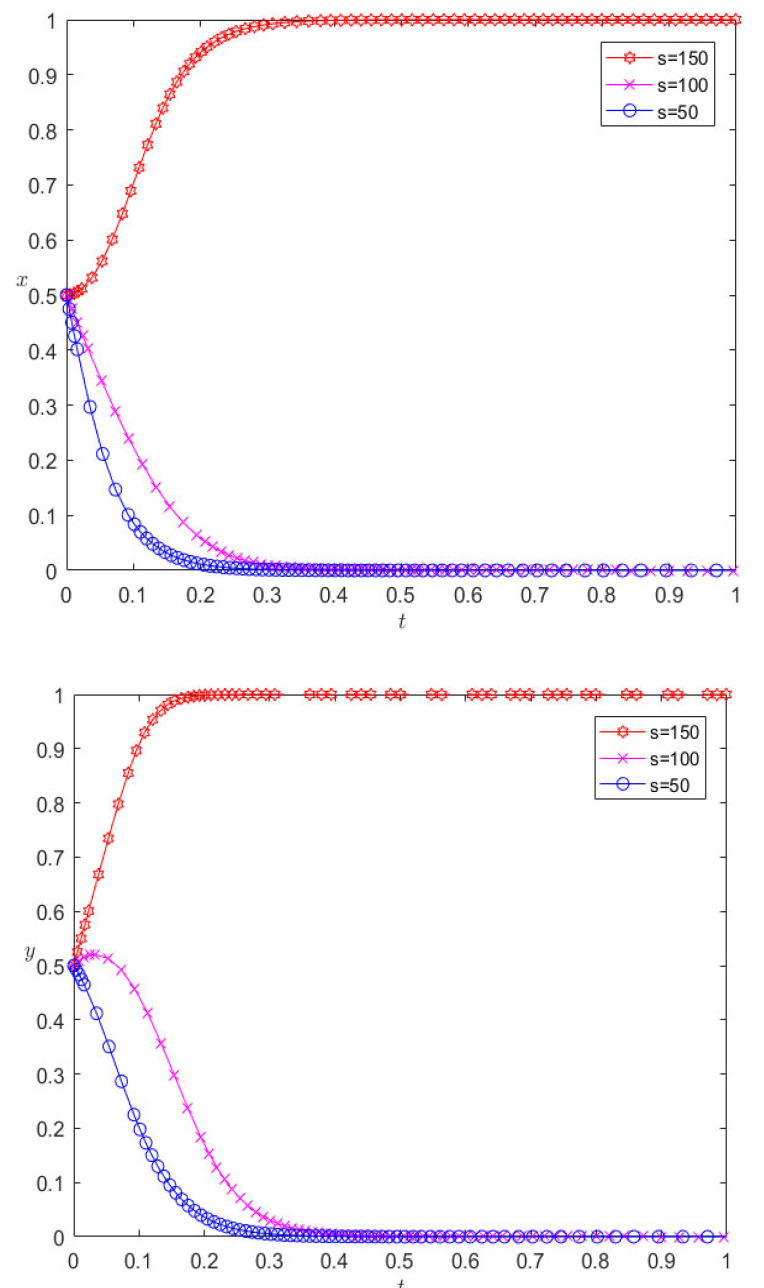
Evolutionary results of reducing the incremental revenue of cooperation.

**Figure 5 ijerph-20-00609-f005:**
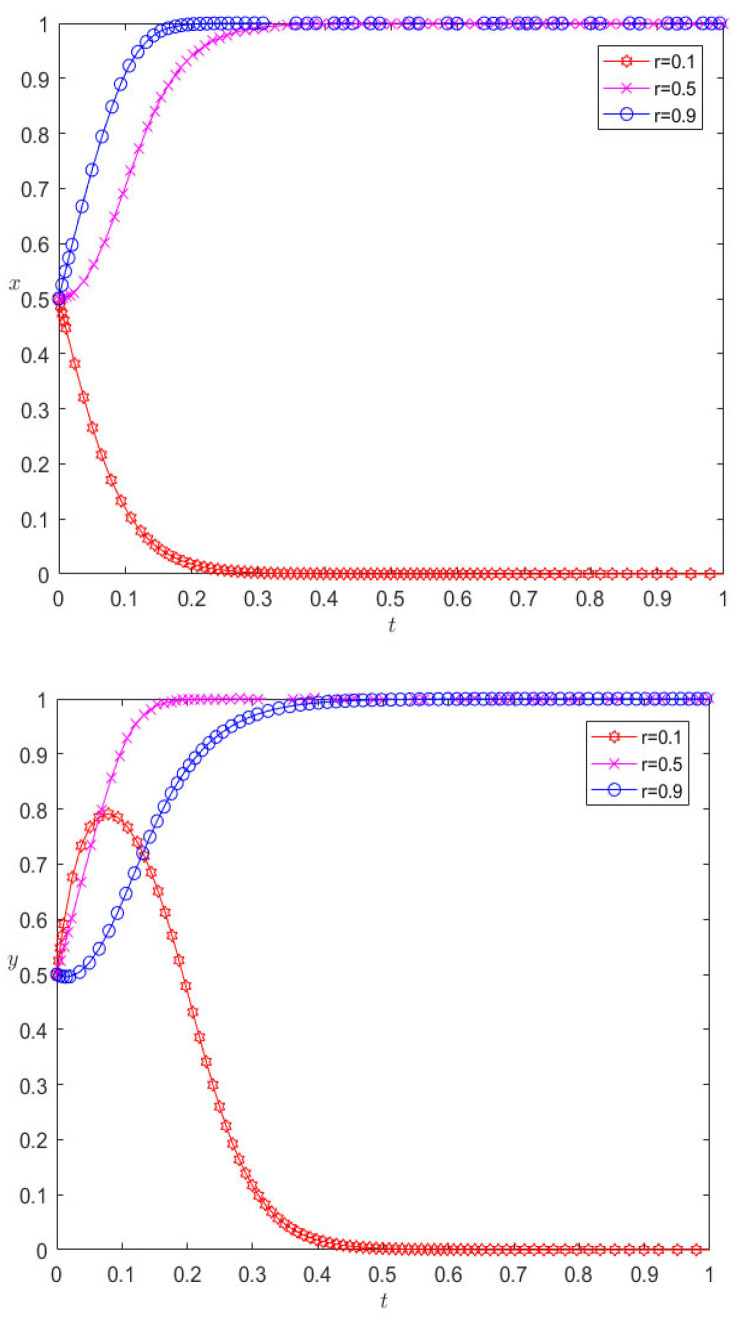
Evolution results of changing the income distribution ratio.

**Figure 6 ijerph-20-00609-f006:**
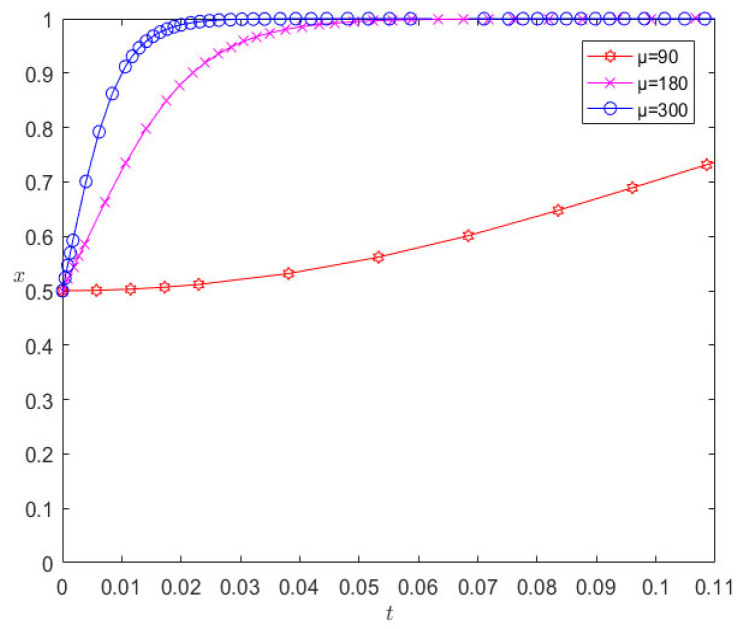
Evolution results of changing liquidated damages.

**Figure 7 ijerph-20-00609-f007:**
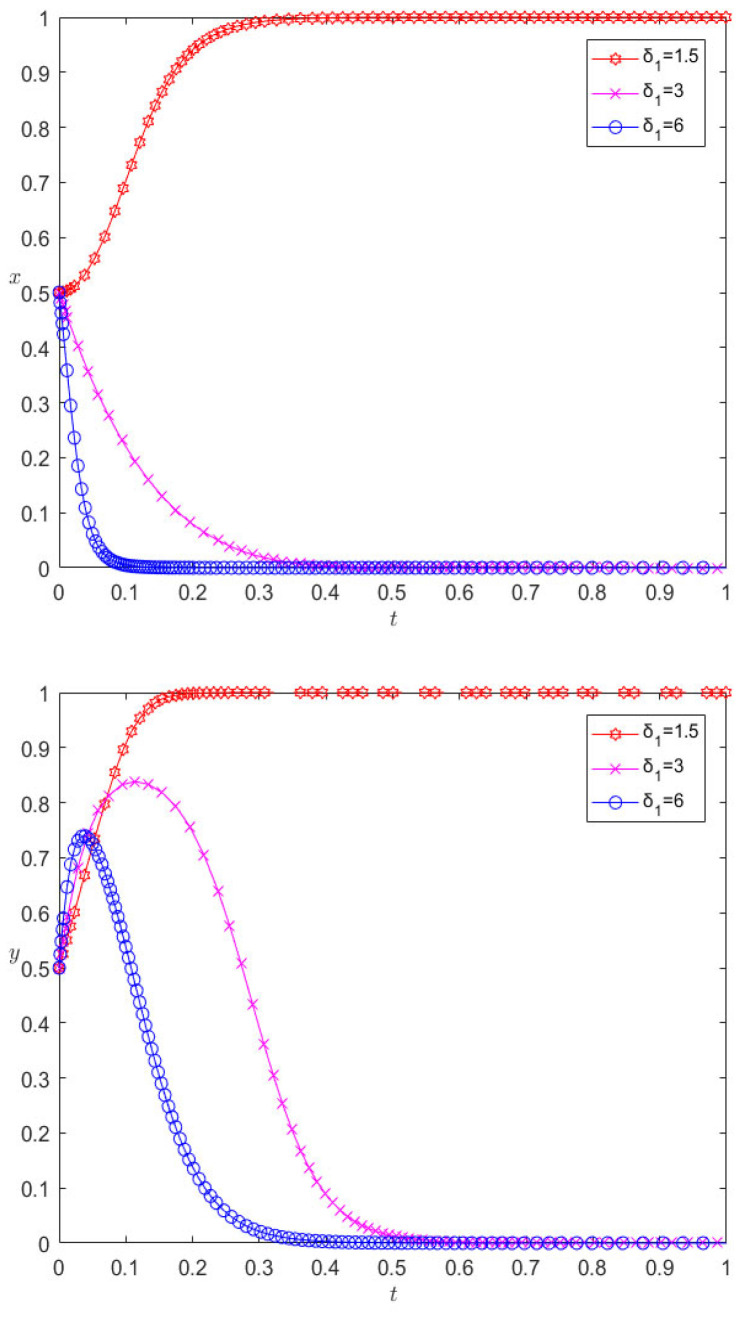
Evolution results of changing excitation parameters.

**Figure 8 ijerph-20-00609-f008:**
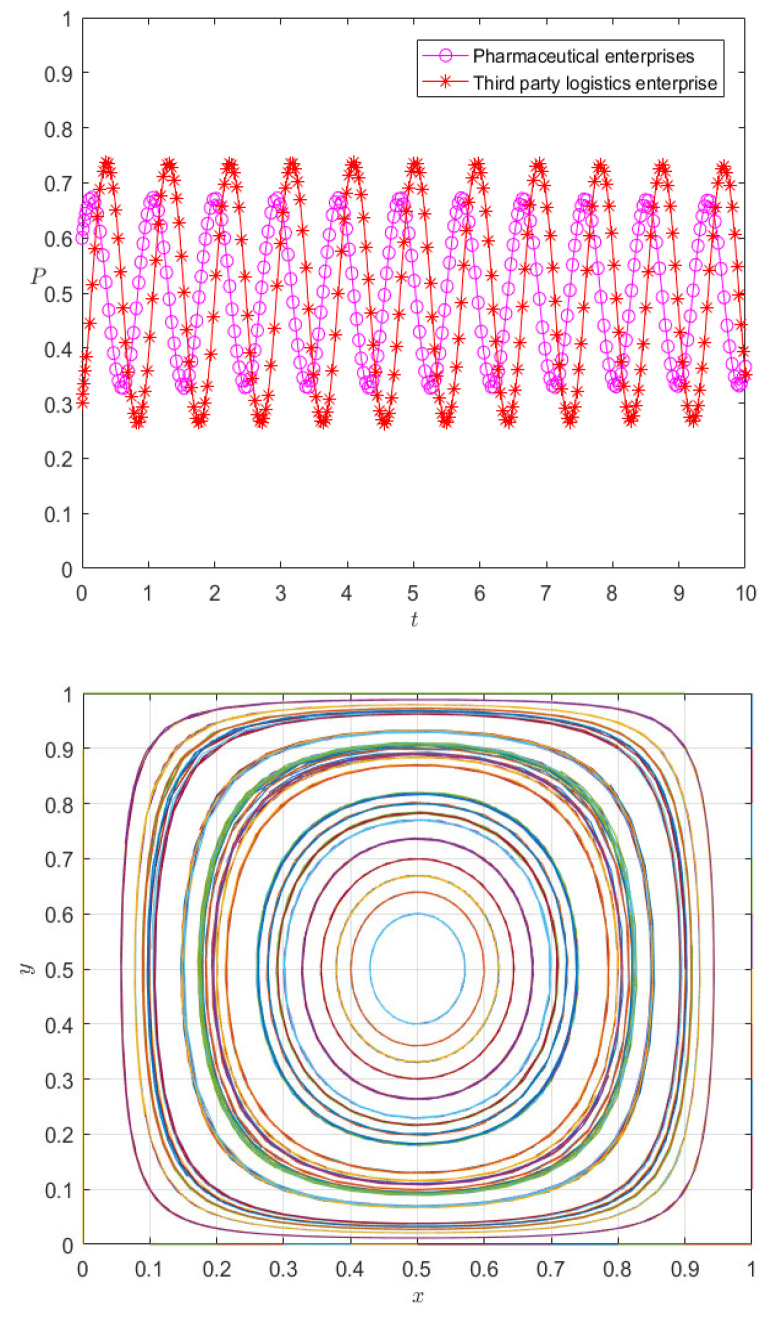
Evolution path diagram of case 3.

**Figure 9 ijerph-20-00609-f009:**
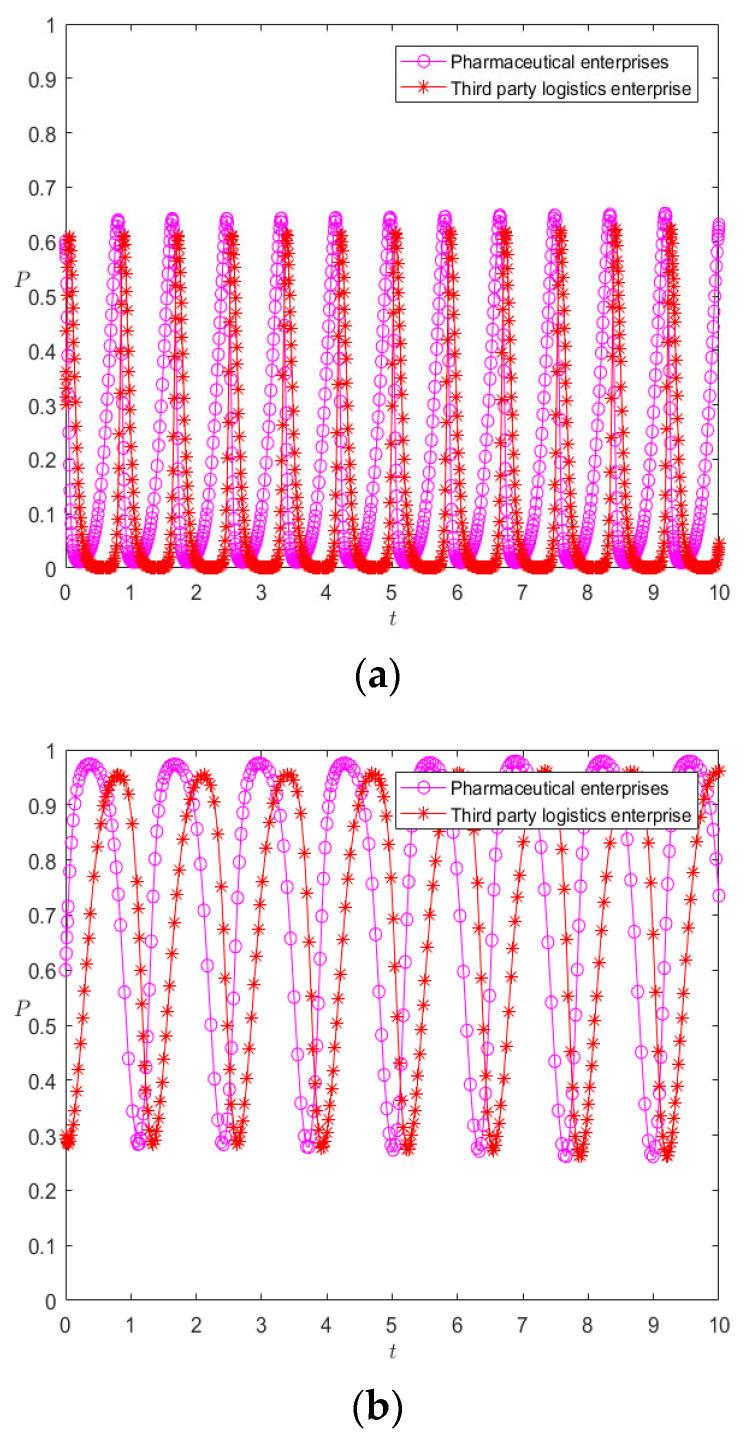
Evolution results of pharmaceutical enterprises (**a**) and 3PL enterprises (**b**) under the static premium and penalty mechanism.

**Figure 10 ijerph-20-00609-f010:**
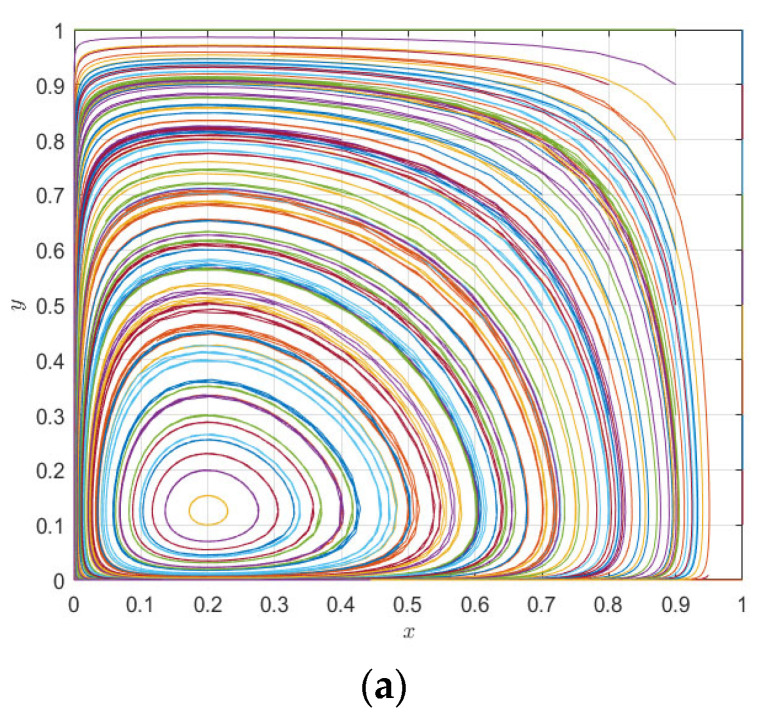
Evolution paths of pharmaceutical enterprises (**a**) and 3PL enterprises (**b**) under the static premium and penalty mechanism.

**Figure 11 ijerph-20-00609-f011:**
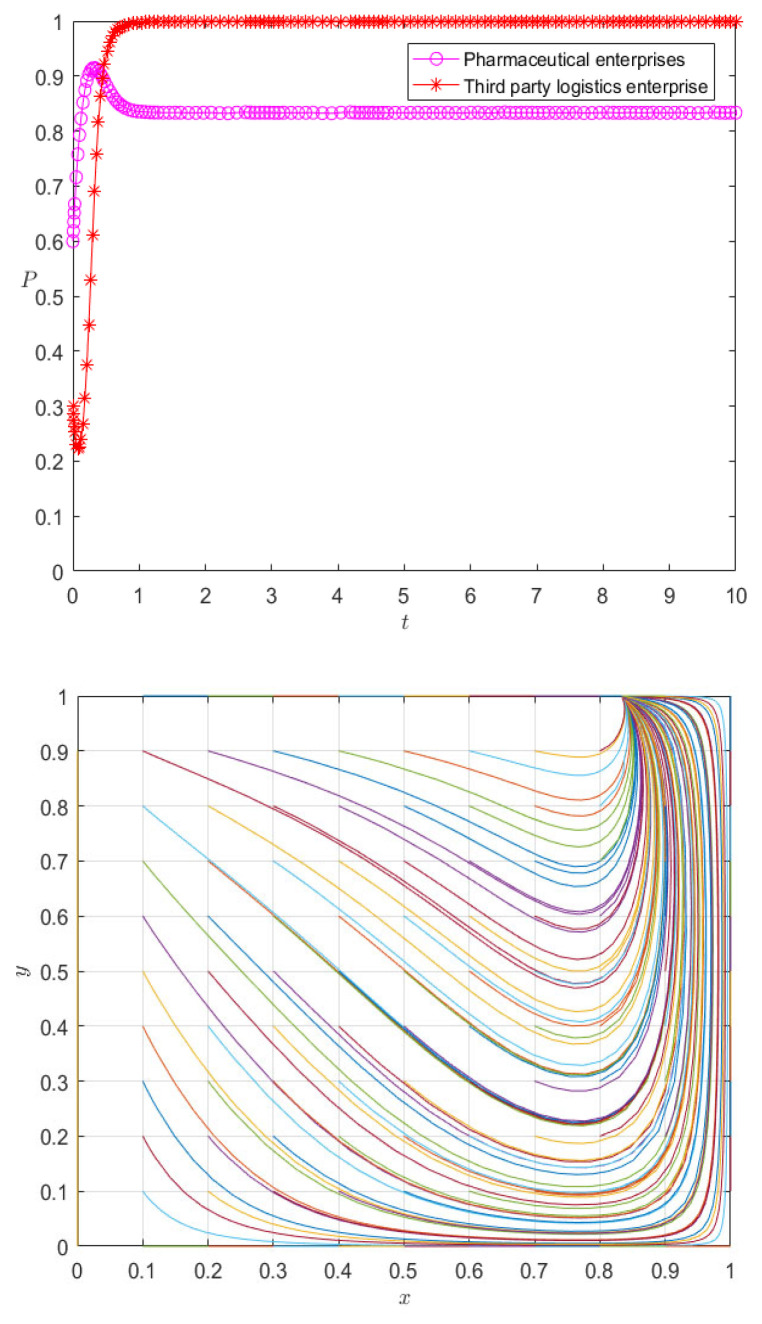
Evolution paths of pharmaceutical enterprises and 3PL enterprises under the dynamic premium and penalty mechanism.

**Table 1 ijerph-20-00609-t001:** Parameters and definitions.

Parameter	Definition	Range
$P_{m}$	Basic income of pharmaceutical enterprises in logistics distribution	$P_{m}>0$
$P_{l}$	Basic income of 3PL enterprises in logistics distribution	$P_{l}>0$
S	Excess return of cooperation between pharmaceutical enterprises and 3PL enterprises	S>0
e	Cost of maintaining cooperation between pharmaceutical enterprises and 3PL enterprises	e>0
r	Profit distribution ratio of pharmaceutical enterprises in cooperation with 3PL	r∈[0, 1]
$a_{m}$	Competition income of pharmaceutical enterprises when they compete and cooperate with 3PL enterprises	$a_{m}>0$
μ	Cost of competition default of pharmaceutical enterprises or 3PL enterprises	μ>0
$\eta_{l}$	Third-party logistics enterprises bear the cost of cooperation maintenance when pharmaceutical enterprises compete and 3PLenterprises cooperate	$\eta_{l}>0$
$a_{l}$	The 3PL enterprises’ competitive income when pharmaceutical enterprises cooperate and 3PL enterprises compete	$a_{l}>0$
$\eta_{m}$	The 3PL bear the cost of cooperation maintenance when pharmaceutical enterprises cooperate and 3PL enterprises compete	$\eta_{m}>0$
$\delta_{1}$	Incentive coefficient of pharmaceutical enterprises to 3PL enterprises	$\delta_{1}>1$
R	Incentive cost of pharmaceutical enterprises to 3PL enterprises	R>0
$\delta_{2}$	Restriction coefficient for pharmaceutical enterprises to compete with 3PL enterprises	$\delta_{2}>1$
T	Restriction cost of pharmaceutical enterprises to 3PL enterprises	T>0
x	Probability of pharmaceutical enterprises choosing cooperation	x∈[0, 1]
1−x	Probability of pharmaceutical enterprises choosing competition	1−x∈[0, 1]
y	Probability of 3PL enterprises choosing cooperation	y∈[0, 1]
1−y	Probability of 3PL enterprises choosing competition	1−y∈[0, 1]
$E_{A1}$	Expected benefits of pharmaceutical enterprises cooperation	$E_{A1}\in \left[-\infty ,+\infty \right]$
$E_{A2}$	Expected benefits of pharmaceutical enterprises competition	$E_{A2}\in \left[-\infty ,+\infty \right]$
$E_{A}$	Average income of pharmaceutical enterprises adopting mixed strategy	$E_{A}\in \left[-\infty ,+\infty \right]$
$E_{B1}$	Expected benefits of 3PL enterprises to choose cooperation	$E_{B1}\in \left[-\infty ,+\infty \right]$
$E_{B2}$	Expected benefits of 3PL enterprises to choose competition	$E_{B2}\in \left[-\infty ,+\infty \right]$
$E_{B}$	Average income of 3PL enterprises adopting mixed strategy	$E_{B}\in \left[-\infty ,+\infty \right]$
F(x)	Replication dynamic equation for pharmaceutical enterprises to choose cooperation strategies	F(x)∈[−∞,+∞]
F(y)	Replication dynamic equation for 3PL enterprises to choose cooperation strategies	F(y)∈[−∞,+∞]

**Table 2 ijerph-20-00609-t002:** Income matrix of co-opetition of medical product distribution in county.

Pharmaceutical Enterprise	3PL Enterprise	
	Cooperation (y)	Competition (1−y)
cooperation (x)	$P_{m}+r\left(s-e\right)-\delta_{1}R$, $P_{l}+\left(1-r\right)\left(s-e\right)+\delta_{1}R$	$P_{m}+\mu -e-\eta_{m}+\delta_{2}T,$ $P_{l}+a_{l}-\mu -\delta_{2}T$
competition (1−x)	$P_{m}+a_{m}-\mu$, $P_{l}+\mu -e-\eta_{l}$	$P_{m},$ $P_{l}$

**Table 3 ijerph-20-00609-t003:** Determinants and traces of equilibrium points.

Equilibrium Point	det(J)	tr(J)
O(0, 0)	$\left(\mu -e-n_{m}+\delta_{2}T\right)*\left(\mu -e-n_{l}\right)$	$\left(\mu -e-n_{m}+\delta_{2}T\right)+\left(\mu -e-n_{l}\right)$
A(1, 0)	$\left(e-\mu +n_{m}-\delta_{2}T\right)*$ $\left(\mu -a_{l}+\delta_{1}R-\delta_{2}T+\left(e-s\right)\left(r-1\right)\right)$	$\left(e-\mu +n_{m}-\delta_{2}T\right)+$ $\left(\mu -a_{l}+\delta_{1}R-\delta_{2}T+\left(e-s\right)\left(r-1\right)\right)$
B(0, 1)	$\left(\mu -a_{m}-\delta_{1}R-r\left(e-s\right)\right)\left(e-\mu +n_{l}\right)$	$\left(\mu -a_{m}-\delta_{1}R-r\left(e-s\right)\right)+\left(e-\mu +n_{l}\right)$
C(1, 1)	$-\left(\mu -a_{m}-\delta_{1}R-r\left(e-s\right)\right)*$ $\left[-\left(\mu -a_{l}+\delta_{1}R-\delta_{2}T+\left(e-s\right)\left(r-1\right)\right)\right]$	$-\left(\mu -a_{m}-\delta_{1}R-r\left(e-s\right)\right)$ $-[\left(\mu -a_{l}+\delta_{1}R-\delta_{2}T+\left(e-s\right)\left(r-1\right)\right)$
$D\left(x^{*},y^{*}\right)$	∗	0

## Data Availability

All data are contained within the article.

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
