# Peer review of "Research on Co-Opetition Mechanism between Pharmaceutical Enterprises and Third-Party Logistics in Drug Distribution of Medical Community"

_ijerph, 2022, doi:10.3390/ijerph20010609_

Round 1
Reviewer 1 Report
This is an interesting and original article. The article presents the model of competition and cooperation between pharmaceutical companies and 3PL companies in the field of drug distribution. The article uses the Jacobian matrix and numerical Matlab simulation. The article is in order. The conclusions are substantiated and supported by the results. The results provide advancement in current knowledge in China.
The article can be corrected: line 517: Matalb - should be Matlab.
Reviewer 2 Report
The graphic part of the study is a bit lacking in legibility. There are a lot of colored lines in the drawings that are not described. It looks nice, but it's hard to understand what's going on. If the accuracy of the graphs can't be improved, it might be worth describing them more clearly in the text.
It would be useful to relate the simulation results and conclusions to specific situations in the pharmaceutical industry to show their practical application. The work would then gain in utilitarianism.
Reviewer 3 Report
This manuscript focuses on an interesting problem that the co-opetition mechanism between pharmaceutical enterprises and third-party logistics in drug distribution of medical community, explores the game strategy choice between pharmaceutical enterprises and 3PL for the solution of drug distribution under the condition of information asymmetry.
The review comments are as follows:
Comment 1. The relevant policies will have an important influence on drug distribution. The authors should reinforce the latest relevant policies in the second paragraph of the introduction. So as to provide the latest policy basis for this study.
Comment 2. In order to enhance the readability of this manuscript, the authors should clearly highlight their academic contribution for this study in the third paragraph of the introduction.
Comment 3. This manuscript has a good many of equations and parameters. So the authors should give a centralized description of relevant parameters in the first part of model construction.
Comment 4. The discussion of model analysis and numerical simulation conforms to the specification. However, the authors should strengthen the practical significance interpretation of the research results.
Comment 5. Check the full text carefully to avoid typos and incoherent sentences.
